# Developments and Prospects of Farmland Application of Biogas Slurry in China—A Review

**DOI:** 10.3390/microorganisms11112675

**Published:** 2023-10-31

**Authors:** Zichen Wang, Isaac A. Sanusi, Jidong Wang, Xiaomei Ye, Evariste B. Gueguim Kana, Ademola O. Olaniran, Hongbo Shao

**Affiliations:** 1Institute of Agricultural Resources and Environment, Jiangsu Academy of Agricultural Sciences, Nanjing 210014, China; 20160028@jaas.ac.cn (Z.W.); shaohongbochu@126.com (H.S.); 2Discipline of Microbiology, School of Life Sciences, University of KwaZulu-Natal, Pietermaritzburg 4000, South Africa; sanusi_isaac@yahoo.com (I.A.S.); olanirana@ukzn.ac.za (A.O.O.); 3Key Laboratory of Crop and Livestock Integrated Farming, Ministry of Agriculture and Rural Affairs, Nanjing 210014, China; yexiaomei610@126.com; 4Key Laboratory of Saline-Alkali Soil Improvement and Utilization (Coastal Saline-Alkali Lands), Ministry of Agriculture and Rural Affairs, Liuhe Observation and Experimental Station of National Agricultural Environment, Nanjing 210014, China

**Keywords:** biogas slurry, liquid digestate, anaerobic digestion effluent, microorganisms, harsh environments, plant–soil interactions, environmental stress

## Abstract

Biogas slurry (BS) is an attractive agricultural waste resource which can be used to regulate soil microbial communities, enhance nutrient absorption capacity of crops, promote plant–soil interactions, and consequently, increase crop productivity. Presently, BS discharge is not environmentally friendly. It is therefore necessary to explore alternative efficient utilization of BS. The use of BS as fertilizer meets the requirements for sustainable and eco-friendly development in agriculture, but this has not been fully actualized. Hence, this paper reviewed the advantages of using BS in farmland as soil fertilization for the improvement of crop production and quality. This review also highlighted the potential of BS for the prevention and control of soil acidification, salinization, as well as improve microbial structure and soil enzyme activity. Moreover, this review reports on the current techniques, application methods, relevant engineering measures, environmental benefits, challenges, and prospects associated with BS utilization. Lastly, additional research efforts require for optimal utilization of BS in farmlands were elucidated.

## 1. Introduction

Biogas slurry (BS) is one of the by-products of anaerobic fermentation [1,2] of biodegradable organic wastes such as livestock and poultry manure, agricultural and forestry wastes, human excrement, urine, and kitchen waste [3]. These wastes undergo anaerobic fermentation in a closed container to produce methane, carbon dioxide and other residues [4]. The by-products of anaerobic fermentation are mainly composed of the solid matter (biogas residue) and the liquid matter (biogas slurry): liquid digestate, or anaerobic digestion effluent [5]. BS has complex components [6,7], which are rich in nitrogen, phosphorus, potassium, copper, iron, zinc, manganese, amino acids, organic acids, hydrolases, vitamins, and other components that are beneficial to plant growth and development [2,8,9,10]. It also contains substances such as 8-hydroxy-3,4-dihydroquinoline-2-ketone and 3,4-dihydroquinoline-2-ketone that have inhibitory effects on pests and diseases [11,12,13]. Moreover, BS contains heavy metal components such as mercury, cadmium, chromium, arsenic, and lead, which are harmful to human beings [14,15]. The global production of BS has exceeded 2 billion tons/annual of which China’s annual production of BS has exceeded 1.12 billion tons [7,16,17]. Improper disposal of BS will adversely affect the soil, water, air, and consequently, plants, animals, and microorganisms in these environments. Unsafe disposal of BS has become a problem that must be solved by a sustainable approach such as usage as soil manure. The urgency of the situation has attracted the attention of both governments and scientists. This will help in the management of agricultural source waste pollution.

BS can be treated before being released to the environment using traditional sewage treatment processes such as oxidation pond method [18,19,20], artificial wetland method [21,22], activated sludge method [23,24], membrane concentration method [25,26,27,28], and chemical flocculation method [29,30]. These techniques are faced with the challenge of waste disposal and high operational cost [31]. For the direct treatment of BS, the resource utilization of BS is more in line with the requirements of sustainable development and green environment [32,33]. Soil fertilization is a major way of BS utilization [34,35], which can improve soil structure, adjust the proportion of various nutrients in soil, regulate soil microbial communities and functions, and enhance the balanced nutrient absorption capacity of crops, thereby promoting plant–soil interactions and overall increasing crop productivity [36,37,38,39]. Current reports show that BS discharged from intensive large-scale livestock and poultry farms and biogas plants has exceeded the carrying capacity of adjacent farmland. Similarly, BS discharged is greatly affected by agricultural seasonality. So it is particularly necessary to expand diversified treatment and utilization [40,41]. This requires an in-depth understanding of the status quo and development trend of BS utilization. However, the existing literature lacks a comprehensive overview of the progress of BS utilization. Hence, this paper using systematic approach, overview of online sources, peer-reviewed articles, and published books review the advantages, approaches, application methods, and challenges of using BS in farmland, in order to provide a reference for the development and utilization of BS resources and the development of new ways of eco-friendly and efficient BS application in farmland. In this regard, over 750 research documents on biogas slurry in China in the past 20 years from January 2000 to December 2019 were reviewed.

## 2. Composition of Biogas Slurry

The physicochemical properties of BS are closely related to raw materials, anaerobic fermentation technological process and storage methods [42]. The nutrient elements, heavy metals, antibiotics, and other residual substance content of different types of BS is shown in Table 1 [9,10,15,43,44]. Other research data show that the water content of BS is as high as 95% or more [45], with a weak alkalinity. Moreover, the amino acid, vitamin, and plant hormone content in BS is shown in Table 2 [9,12]. Since the composition of BS varies greatly, the composition of BS should be determined before its application, so as to achieve reasonable safe decomposition.

## 3. Advance in Farmland Consumption of Biogas Slurry

### 3.1. Advantages of Using Biogas Slurry in Farmland

#### 3.1.1. Soil Fertilization

Farmland application of BS can improve the physical and chemical properties of farmland soil [38,46,47] while effectively valorizing the BS [48]. This has a direct positive effect on increasing soil organic matter, improving soil structure and maintaining soil fertility [49,50,51]. The decrease in soil organic matter content is one of the reasons for the deterioration of soil structure and the reduction in soil productivity [52]. The application of BS rich in organic matter to farmland can increase the content of organic matter, especially dissolved organic matter in the soil, thereby improving soil structure [53,54,55]. For instance, the pig manure BS can increase the organic matter content of the topsoil to 3.0 kg/hm^2^ [56]. After 5 years of BS irrigation, the soil organic carbon content increased significantly by 90.3% compared with the soil with chemical fertilizers [57]. However, some studies applying BS from chicken manure, pig manure, and cow manure on the soil in comparison to the control had no significant effect on soil organic matter content [58]. Also, BS rich in nitrogen promotes the consumption of organic carbon by non-autotrophic microorganisms [59], thereby offsetting the accumulation of organic matter present BS in the soil [60,61].

The effect of BS application on soil organic matter content is related to the application mode and the composition of the BS. The increase in soil organic matter content was proportional to the amount of BS applied [62]. The soil organic matter content of all the different fertilizers applied gradually decreased with the growth of corn, while the BS treatment was the opposite. At the mature stage, the organic matter content of all the treatments with BS was found to be significantly increased [63].

The consumption of BS on farmland can enhance soil permeability, water retention, and fertilizer retention capabilities, an advantage that chemical fertilizers do not have [64]. For instance, the 3-year application of BS (165.1 and 182.1 t/hm^2^) improved the nutrient content of yellow soil under rice–rape rotation and promoted the formation of soil aggregate structure [65]. With the increase in the amount of BS in the mixed solution, the soil stability indicators of dry-fed red soil aggregates (i.e., soil > 0.25 mm water-stable aggregate content, aggregate mean mass diameter, and geometric mean diameter) showed an upward trend, while the fractal dimension showed a downward trend. Similarly, after long-term BS irrigation, soil porosity, soil aggregate structure, and microorganisms in soil increased [66].

Similarly, the application of BS can effectively adjust the proportion of various nutrients in the soil, with the potential to enhance the nutrient absorption capacity of crops, increase crop resistance to diseases [37], increase soil organic matter and improve soil structure [36]. For example, soil ammonium nitrogen and soil nitrate increased by 47.8% and 19.0% when treated with BS compared with the control, respectively [48]. Also, after applying BS formulated fertilizer in the orchard, the soil organic matter content in each soil layer increased from 3.0% to 3.9%, total phosphorus increased from 5.6% to 18.6%, and the available potassium increased from 25.2% to 39.2% [67]. Also, compared with the control without any addition of BS, the BS from chicken manure, pig manure, and cow manure added to soil under equal nitrogen conditions improved inorganic nitrogen, total nitrogen, total phosphorus, available potassium, pH, and conductivity. It is worth noting that the increase in soil nitrate nitrogen is optimal after pig manure BS treatment, followed by chicken manure BS treatment [58,68].

Likewise, continuous application of BS can increase the content of total nitrogen, total potassium, and available nitrogen in farmland soil [69,70]. The content of soil organic matter, cation exchange capacity, electrical conductivity, soil total nitrogen, total phosphorus, total potassium, alkali-hydrolyzed nitrogen, available phosphorus, available potassium, and NH_4_^+^-N content of a paddy field with continuous application of BS for four years were significantly higher than those without BS application [71]. Interestingly, a tea garden with extremely low soil fertility level reached high fertility level after continuous application of BS for 2 years. The soil indexes of the soil treated with BS for 4 years were significantly improved compared with the soil without BS treatment [72].

#### 3.1.2. Improvement in Crop Production

Achieving increase in crop yield is the primary objective soil fertilization such as using BS as fertilizer (see Table 3). Using meta-analysis method, the effect of BS application on crop yield under different conditions was quantitatively analyzed [73]. The results showed that the effect of BS application on wheat, corn, tomato, and rice yield were all improved. Moreover, the impact of BS on farmland could be influenced by soil type or prevailing climate conditions. For example, BS application in northwest and north China increased crop yield significantly compared with other regions such as southwest and east China regions.

The physicochemical properties, application period and concentration of BS plays an important role in formulating a safe and efficient use of BS [50,65,109]. Applying 50% BS instead of chemical fertilizer resulted in the same corn yield as using chemical fertilizer only [110]. Similarly, when applying BS derived from anaerobic fermentation of pig urine and feces, a significant increase in the yield of corn was obtained. When the concentration was controlled within the range of 60–90 t/hm^2^, the maximum corn yield was obtained [81]. Likewise, many studies have shown that the application of BS was beneficial in rice cultivation to increase rice yield [61,74,77,111]. On the other hand, it was reported that the complete replacement of fertilizer with BS significantly reduces rice yield [40,112]. Hence, the application of appropriate BS is critical in improving crop yield than conventional fertilization [78].

#### 3.1.3. Quality Improvement

Moreover, the application of BS in farmland can improve the nutritional quality of cultivated crops. This will turn enhance the commodity attributes and economic value of crops [82,111]. For instance, the increasing use of BS in the irrigation of rapeseed cultivation improved Fe, Mn, Cu, and Zn mineral content in rapeseed, while the content of oleic acid, Ca, and Mg in rapeseed increased first and then decreased. The optimal quality of rape was achieved when BS was applied in the range of 78.8–101.3 t/hm^2^ [113]. Though the application of BS will increase the protein content of rice and improve the nutritional quality of rice [77,111], some studies have also shown the application of BS has little impact on rice nutritional quality [114,115]. Based on the 8–9-year data analysis of long-term BS application, an improvement in the rice yield, taste value of rice, and gel consistency of rice was obtained when compared with those obtained from chemical fertilizer treated soil [80]. The application of BS promoted the accumulation of vital components such as polysaccharides, carotenoids, flavonoids, and betaine in lycium barbarum fruit, thereby improving the nutritional quality of lycium barbarum and its efficacy [116]. Similarly, the application of nitrogen fertilizer and fermented pig manure in the cultivation of Chinese cabbage showed lower content of amino acids and soluble sugar when compared with the application of pig manure BS alone [117]. Likewise, in a related study, the effects of BS treatment with concentration of 25%, 50%, 75%, and original liquid on the quality of *Capsicum* spp. were studied. The results showed that with the increase in BS concentration, the chlorophyll content, vitamin C content, soluble sugar content, and organic acid content of *Capsicum* spp. were improved [118]. It was also found that the application of BS to replace chemical fertilizer could significantly increase the soluble sugar, soluble solids and sugar acid ratio of muskmelon [119].

#### 3.1.4. Bacteriostatic

Biogas slurry undergoes long-term anaerobic fermentation to produce a variety of biologically active substances, such as organic acids, vitamin B12, and gibberellin, which could inhibit the proliferation of soil bacteria, fungi, and viruses [12]. In addition, the high concentration of NH_4_^+^-N in the BS has the potential of killing pests and pathogenic bacteria [120,121]. For instance, fresh BS from cattle farm has strong inhibition effect on *botrytis cinerea*, *phytophthora capsici*, *alternaria solani*, *colletotrichum gloeosporioides*, *botrytis capsici*, and *botrytis cinerea* of eggplant. However, when the BS storage time was increased, the inhibition rate of BS against *phytophthora capsici* and *fusarium solani* decreased significantly [122]. Similarly, concentrated BS remarkably inhibits the growth of cotton verticillium wilt mycelium (the inhibitory effect of 0.5% concentrated solution BS on cotton verticillium wilt disease was 64.9%). The BS also prevent spore production, conidial germination and microsclerotia germination [123]. Likewise, the application of BS had effective repellent effect on adult brown rice plant hopper [101]. The spraying of biogas slurry with 66.6% concentration had the best repellent effect [106].

Furthermore, the application of BS was effective in the prevention and control of root borne diseases of crops [124,125,126,127]. Although nitrogen input is considered to be the key factor to stimulate soil microbial biomass carbon [46], a large amount of ammonium nitrogen in BS may play a role in inhibiting microbial growth in the short term. The bacteria population in soil decreased after BS was applied in pot culture system [120,121]. For example, irrigation with high concentration of BS in broccoli field reduces soil fungi by 55.0% [128], thus significantly reducing the plant disease index. Similarly, when BS was applied to watermelon, a substantial inhibitory effect on *Fusarium* wilt was observed, and the disease index was lowered by 36.4% compared to the control treatment [125,126]. Further analysis showed that the inhibition of basidiomycota and mortierella growth was the reason for the decrease in the disease index [39]. Remarkably, root irrigation with BS effectively prevents and cure astragalus root rot. The same inhibitory effect was obtained when this was repeated many times [129]. Likewise the inhibitory effect of 1.3% BS concentrate on cotton verticillium wilt by root irrigation reached 78.0% [123].

#### 3.1.5. Prevention and Control of Soil Acidification

Soil acidification is one of the main factors affecting agricultural productivity as well as negatively impacting the environment. Soil acidification will destroy the structure of biological cell membranes, reduce microbial activity of soil microorganisms, and consequently, crop health, growth, and productivity. Prevention and control of soil acidification is of great significance to maintaining sustainable agricultural development [130]. The use of BS in farmland can effectively adjust the proportion of various nutrients in the soil, enhance the ability of soil to buffer acidity and alkalinity changes, reduce the pH value of alkaline soil [91], and increase the pH value of acidic soil, thereby improving soil quality [131,132,133]. Studies have shown that irrigation with BS in coastal poplar forests and coastal saline-alkali rice–wheat rotation fields cause a pH reduction [134,135,136]. For instance, the soil pH of an alkaline paddy field treated with BS for 4 years was significantly lower than that of soil without BS application [71]. Compared with conventional fertilization treatment, BS application can effectively prevent further soil acidification caused by long-term application of chemical fertilizers [80]. Similarly, a 3-year field experiment carried out on the yellow soil under rice–rape rotation showed that the application of BS (165.1 and 182.1 t/hm^2^) could increase the soil pH [65]. Lower concentration of BS do not prevent soil acidification, while higher BS concentration inhibited the growth of acidobacteria, thereby reducing soil acidification [39]. Likewise, long-term application of BS, resulted in an increasing trend of the soil pH of cultivated Hongmeiren citrus. The soil pH after 4 years of BS application was significantly higher than that of the conventional fertilization [108]. Also, the application of BS in an economic fruit plantation such as a tea garden [137], grapefruit [55], apple [97], and citrus [107] showed similar results of an increasing soil pH value.

#### 3.1.6. Improved Microbial Structure and Soil Enzyme Activity

The role of soil organisms in underground ecological processes are vital to maintaining a healthy farmland fertility and productivity [138]. An important group of soil organisms are the microorganisms; as decomposers in the food web [139], they occupy more than 80% of food web biomass [68]. These microbes participate in the decomposition and synthesis of soil organic matter, the fixation and release of nutrients, as well as the degradation of pollutants. The impact of farmland application of BS on underground ecological processes will inevitably lead to changes in microbial community structure, metabolic characteristics, and functional diversity, which in turn can be used as important indicators for the evaluation of the health of farmland.

Soil microbial biomass C/N ratio reflects the composition of soil microbial flora. The lower the microbial biomass C/N ratio, the more bacteria in the soil. The application of BS can increase the culturable quantity of soil bacteria [140], fungi [38], and actinomycetes [141] to a certain extent. In a related study, after BS application, the ratio of soil microbial biomass C/N decreased by 25.2–48.0% [108]. The application of BS promoted the proliferation of soil bacteria, and the activity of soil bacteria increased significantly with long term application of BS on farmland [142]. The ratio of bacteria and fungi (B/F) in the soil is usually used to evaluate the soil microbial flora [143]. A high B/F value indicates that the soil is a “bacterial type” with high fertility and less damage to the soil, while a low B/F value indicates that the soil is a “fungal type” with low fertility and high damage to the soil. For instance, the treatment of BS mixed with chemical fertilizer reduced the B/F value. The increase in the concentration of BS application resulted in a B/F value that initially decreased and then increased, while the application of pure BS increased the B/F value considerably [144]. Hence, the B/F value of soil can be kept stable or even increased by using appropriate BS, and consequently, improving the soil fertility.

Nitrogen, phosphorus, potassium, organic matter, growth hormone, humic acid, cellulose, and other substances in BS can further promote the growth and enrichment of soil dominant bacteria [145] as well as promote microbial alpha diversity by improving soil structure and increasing organic matter [49,121,125,126,146,147]. For example, in paddy field with BS applied continuously for 6 years, campylobacter, proteus, and acidobacter were the dominant bacteria, which shows that BS can improve soil microbial structure and, consequently, soil quality and soil fertility [148]. Moreover, the increase in BS concentration resulted in the actinomycetes population; however, excessive BS application inhibits the growth of actinomycetes [39,65]. The Chao1 index and Shannon index of soil bacteria treated with 180 t/hm^2^ BS were higher than those of control treatments; however, the Chao1 index of fungi was lower than that of chemical fertilizer (100 t/hm^2^ and 220 t/hm^2^ treatments). The concentration of BS at 180 t/hm^2^ can improve the bacteria richness and diversity, while reducing the diversity of fungi [39].

In addition, the application of BS in farmland has a certain impact on the activities of soil organisms. When the concentration of BS increased from 0 to 300 m^3^/hm^2^, the density of soil organisms increased by 94%, the number of the groups increased by about 2, and the dominance index increased by 9.4% (*p* < 0.05). When 66% of BS was used to replace chemical fertilizer, soil organism density, number of groups, and dominance index were at the highest. The principal component analysis of the application of BS alone or mixed with the chemical fertilizer, showed collembola, prestoma, and ortychia were the most sensitive groups, and they could be used as indicators of the response of small arthropods in the soil to decomposing BS.

Moreover, proper application of BS in farmland can increase soil enzyme activity (see Table 4) thereby improving soil carbon and nitrogen transformation. Studies have shown that the application of BS increased the activities of soil phosphatase, protease, dehydrogenase, sucrase, catalase, and urease. After 4 years of BS application, the activities of the six enzymes were significantly higher than those of the control [108].

### 3.2. Approaches of Using Biogas Slurry in Farmland

#### 3.2.1. Seed Soaking

The abundance nitrogen, phosphorus, potassium, various trace elements, growth hormones, and other substances in BS can be absorbed and utilized by seeds through seed soaking and infiltration. This can accelerate the metabolism of seeds during the dormant period, thereby promoting seed germination. Reports have shown that the proper concentration of BS and soaking time could improve the germination rate of seeds and promote the growth of seedlings. Soaking seeds with 50% BS for 5 h had the best comprehensive effect on the germination of marigold seeds and seedling growth [152]. For instance, the germination rate and seedling rate of watermelon seeds treated with 40% BS for 24 h was observed to give the best performance [86]. Similarly, soaking seeds with 25% BS for 5 h had a significant effect on seed germination and seedling growth of Astragalus mongholicus [153]. In addition, soaking seeds with BS can increase crop yield. The seed soaking treatment of wheat seeds with BS at optimal exposure time can increase the germination rate by about 13% compared with the water treatment. This resulted in the seeds emerging 3 days earlier, the leaf length increasing by 1.70 cm, the leaf width increasing by 0.10 cm, the dry weight of seedlings increasing by 0.70 g, the maturity period shortening by 2 days, and the yield per hectare increasing by 379.50 kg [154]. Hence, BS could be a potential fertilizer for improve agricultural productivity and sustainable green environment. The effect of soaking seeds with biogas slurry has been reported by many studies, but there is a lack of systematic understanding of the concentration, time, temperature, and operational precautions during the soaking process for different crop seeds. The corresponding mechanism of the soaking effect of biogas slurry still needs further research, and as one of the ways to utilize biogas slurry, its environmental and economic benefits need to be evaluated.

#### 3.2.2. Foliar Fertilizer Using Biogas Slurry

Using biogas slurry as a leaf fertilizer is one of the important ways for farmland to utilize BS. BS is often used as foliar fertilizer for it contains a variety of available nutrients and amino acids which can promote plant growth, increase yield, and improve crop quality [54,155]. BS has been directly used as foliar fertilizer to spray on fruit trees and vegetables, which significantly increased chlorophyll content and yield [87,156]. For instance, proper application of BS sprayed on the leaves improved the growth, yield, and quality of tomato plant [105]. Similarly, investigating the effect of BS application on walnut production, Bi T et al. [157] observed that the BS has an enhancing effect on the walnut quality and the control of pests as well as diseases. The authors pointed out that when BS is used as a leaf fertilizer and pest control, it should be diluted and sprayed on the back of leaves.

Likewise, foliar topdressing of BS can increase the yield of cucumber by 6% and tomato by 8% [158]. Adding humic acid and other nutrients to BS up to 10% of the original volume, and then compounding it to organic fertilizer with large, medium, and trace element, as a foliar fertilizer, significantly improve the yield and quality of Chinese cabbage. The yield of Chinese cabbage increased by 23.3%, and the content of vitamin C and soluble sugar increased by 68.5% and 43.1%, respectively. At the same time, soil fertility, enzyme activity, and soil nutrient increased significantly [159]. In addition, topdressing BS application increased *Capsicum* spp. yield [160], and increased the contents of vitamin C, soluble sugar, and protein in *Capsicum* spp., among which the vitamin content increases by 18.3% compared with the market foliar fertilizer [161].

#### 3.2.3. Base Fertilizer Using Biogas Slurry

Biogas slurry as basic fertilizer is the most traditional approach of BS application. Compared with chemical fertilizer, under the same treatment condition as BS as base fertilizer (52.5 t/hm^2^) and root irrigation twice (0.25 kg/root · time), the length, diameter, leaf area, and chlorophyll content of sweet melon vine were increased. Also, in the same study, the weight of melon and melon plant were increased [162]. Similarly, a study by Li and Jiang [163], showed that when the concentration of BS is between 10% and 20%, it is favorable for the growth of the container seedlings of Dendrobium candidum using water moss substrate, while BS between 10% and 30% was found to be favorable for the growth of disk seedlings of Dendrobium candidum using a sawdust pine bark substrate. Under the condition of total application of BS with 600 t/hm^2^ (base fertilizer/top dressing = 1:1), the total panicle number of rice and the yield was increased, while the content of heavy metals in grains did not increase [75]. When the fermented BS of livestock manure is used as the base fertilizer of the tea garden in the autumn and the top dressing in the spring of the following year, the production and quality of spring tea was improved with the content of heavy metals in the soil and tea leaves maintained within a safe range. However, when the BS is applied alone, potassium depletion occurred. Therefore, it is necessary to pay attention to the supplement of potassium in practical application [92]. Practically, the treatment of BS should be carried out according to the ratio of base fertilizer to fruit expanding fertilizer of 1:1. When the application rate of BS was 70–110 t/hm^2^, the plant height and stem diameter of melon were not significantly different from that of compound fertilizer of 600 kg/hm^2^; when the total application of BS is 180 t/hm^2^ (base fertilizer 90 t/hm^2^, fruit expansion fertilizer 90 t/hm^2^), it promoted melon plant growth, dry matter, and fruit quality [39].

#### 3.2.4. Top Dressing Fertilizer Using Biogas Slurry

The application of BS instead of chemical fertilizer for crop top dressing can increase the content of nitrogen and phosphorus in the soil, and could increase with the increase in BS concentration. For instance, the top dressing application of BS containing nitrogen of 396 kg/hm^2^ has a rice yield greater than that treated with chemical fertilizer. Also remarkable is the utilization rate of nitrogen and phosphorus, which was higher in the BS treated soil [112]. Compared with the control (non-BS topdressing treatment), BS topdressing treatment increased the yield of angelica sinensis by 112.89 kg, an increase of 58.0%. Additionally, this significantly reduced the disease index of angelica hemp mouth disease with a preventive effect of 82.3% [164]. Moreover, using the nutrient balance method, it was found that there was no significant difference in the dry matter quality and nutrient content of root, stem, leaf, and fruit of muskmelon between the BS fertilizer integrated topdressing group and the chemical fertilizer group. Hence, the BS fertilizer integration could completely replace the chemical fertilizer [165].

Even though previous studies on the use of biogas slurry as a base fertilizer and topdressing mainly focusing on replacing some chemical fertilizers have been reported, there is a lack of research on the application methods, equipment, relevant engineering measures, parameters, and environmental benefits assessment of BS base and topdressing fertilization application.

#### 3.2.5. Hydroponics

Using BS to replace the inorganic nutrient solution of hydroponic cash crops for vegetable cultivation is one approach of resource utilization of BS. Through a biological floating bed process, celery was hydroponically cultured in different concentrations of BS. After 80 days of planting, the celery which was hydroponically cultured in 30–40-times-diluted BS achieved high environmental and economic benefits [166]. In the concentration range of 3–5%, the stepwise addition and one-time addition of chicken manure BS increased the chlorophyll content, biomass, and vitamin C content of water spinach in the solar greenhouse, while nitrite content was reduced [99]. Compared with ordinary soil cultivation treatment, BS soilless cultivation treatment can significantly increase the number of lateral roots and total yield of water spinach by 45.4% and 12.8%, respectively, as well as reduce nitrate nitrogen content in water spinach by 31.5% [167]. In another related study using BS as nutrient substitute for the second growth stage of lettuce, the replacement of nutrient solution with BS has a better effect on lettuce yield, photosynthetic characteristics, and quality. The replacement ratio of 40% BS has the best effect, and the yield is 67.0% higher than that of the control (lettuce hydroponics with nutrient solution prepared according to the original Yamasaki formula) [168]. After the BS deamination, pretreated and diluted by 5–10 times, lettuce was hydroponically cultured for 35 days. Then, compared with hydroponics in nutrient solution, the relative growth of lettuce increased by 60%, the leaf width became wider by 4–5 cm, the number of leaves increased by 2 pieces on the average, the carotenoids content increased by 20.4%, and the content of nitrate nitrogen improved from 2.1 top 4.0% compared to that of chemical nutrient solution group [169].

Biogas slurry hydroponic microalgae is a new type of resource treatment process with potential and stable operation. Meanwhile, it is also an effective way to achieve high-value utilization of BS. Compared with traditional biochemical methods, it can improve the nitrogen removal efficiency of BS by about 20% [109,170], and obtain higher-efficiency functional microalgae products. Different microalgae were cultured with pig manure-based BS, with the nitrogen removal ability and sugar accumulation potential investigated. *Chlorella vulgaris* ESP-6 showed the best sugar production capacity, with the maximum sugar content and average daily sugar production capacity of 61.5% and 395.73 g/L, respectively. The ammonia nitrogen removal rate and daily average removal concentration were 96.3% and 91.7 mg/L, respectively. Accumulating more carbohydrates in microalgae cells can be regarded as a new strategy for sugar production, which fully proves the value of BS hydroponic microalgae utilization and the regeneration potential of BS waste resources [171].

#### 3.2.6. Animal Feed

The use of BS as animal feed and feed additive is another environmentally friendly approach to comprehensively utilize BS for both ecological and economic benefits. Reports in the literature are mostly found in empirical research and attempts research. For instance, the number of heterotrophic bacteria in the sediments of fish ponds with BS or BS combined with feed was higher than that of cattle dung or BS combined with inorganic fertilizers. Similarly, the sediment–water interaction in fish ponds with BS was better than the conventional fertilized fish ponds [172]. Fish farming with BS can increase the yield and economic benefits of feeding and filter-feeding fish; however, attention should be paid to the amount, the frequency, and the timing of BS dosing [44].

### 3.3. Application Methods of Using Biogas Slurry in Farmland

#### 3.3.1. Drip Irrigation

BS applied in the planting of vegetables, melons, fruits, using drip irrigation has the advantages of uniform application, reduced production cost, promoted nutrient content, and increased yield. Strict filtration and blockage prevention systems need to be put in place in drip irrigation. Compared with spraying BS, drip irrigation with BS could increase the available nutrients in the substrate, as well as improve the growth of crops [100]. Using the method of BS aeration drip irrigation to conduct a plot test on leeks in a greenhouse, it was found that when the concentration of BS was 80%, with aeration coefficient of 1.0, the yield of leeks was the highest up to 230.50 kg/667 m^2^, an increase of 28.5% compared with the control. At the same time, the content of vitamin C in chives increased by 77.8%, while the content of soluble sugar increased by 91.2%, and the content of soluble protein increased by 70.6% [173]. Similarly, the drip irrigation BS treatment in watermelon, cucumber, strawberry, grape, and tomato fruit cultivation achieved a 13.9% increase in watermelon fruit weight [84], 15.7% increase in strawberry fruit weight [174], and 18.1% increase in tomato fruit weight [104]. Similarly, the strawberry yield increased by 18.1% [174], grape yield increased by 18.3% [89], cucumber yield increased by 47% [175], and tomato yield increased in the range of 20.7–59.4% [176,177]. Also noteworthy was the increase in the fruits’ soluble total sugar, Vitamin C, and titratable acid, and the improved fruit firmness [104].

Furthermore, after the BS was diluted with water and was dripped into the saline-alkali soil, the results showed that the soil fertility was significantly improved, the soil pH was decreased, and the desalination effect was significant in the 0–20 cm soil layer, while salt accumulation occurred below the 20 cm soil layer [178]. From the economic point of view, integrated drip irrigation of BS, water, and fertilizer not only greatly reduces the manpower and costs of BS transportation to the fields, but also reduces the application of chemical fertilizers [119]. Compared with the BS flood irrigation treatment, the BS water and fertilizer integrated drip irrigation treatment significantly improved the yield and quality of pear fruit, as well as significant reduction in the soil nitrogen accumulation. Likewise, comparing the use of BS with conventional fertilization, 43% of chemical fertilizer usage can be avoided [88]. The integrated drip irrigation of BS, water, and fertilizer for pear trees also significantly improves the yield and quality of pear fruit, providing an economical and effective fertilization mode for pear trees.

#### 3.3.2. Ditch Irrigation and Flood Irrigation

At present, the methods used to absorb BS in farmland are still mainly furrow irrigation, flood irrigation and surface application, which not only requires a large amount of labor and high labor intensity, but also has large loss of ammonia volatilization. Long-term application may cause secondary soil salinization and accumulation of heavy metals, and increased risk of groundwater contamination [64,179]. Notwithstanding, it was found that the irrigation of BS using the ditch irrigation and flood irrigation approach could increase the root system and the yield of Codonopsis pilosula towards the commercialization [180]. Similarly, the furrow irrigation BS application significantly increased the yield of tomato. The yield increase rate was 10%, and the vitamin C content increased was 1.54 times better compared with the control [181]. In ditch irrigation, when the amount of BS applied was the same, the content of ammonium nitrogen and nitrate nitrogen in the soil treated by furrow application was higher than that of surface application and deep ploughing [182]. Moreover, BS flood irrigation significantly increases the content of soil organic matter, total nitrogen, alkali-hydrolyzed nitrogen, and available potassium.

#### 3.3.3. Spraying Application

The spraying of BS is more common with foliar spraying and soil surface spraying. It is necessary to pay attention to the spraying concentration and spraying amount. Studies have shown that spraying using a 60% concentration of BS or root application of BS, the yield of tomato, radish, celery, and China bean can significantly be increased. Similarly, in celery production, spraying using a 40% concentration of BS has the best yield increase effect [183]. Spraying nectarine leaves with different concentrations of BS can increase the nutrition of nectarine leaves, and at the same time, the single fruit weight, soluble sugar content, and sugar/acid ratio of nectarine fruit were also improved [184]. Foliar spraying of BS can effectively increase the single melon quality of cantaloupe and improve its quality with75% concentration BS spraying having the best effect [93,94]. Likewise, foliar spraying of BS can increase the yield of apple trees and increase the content of vitamin C and soluble sugar in the fruit [185]. The use of BS spray irrigation can improve the soil content of alkali-hydrolyzed nitrogen, available phosphorus, and potassium in the deep soil [133].

#### 3.3.4. Combine Application

The application of BS is no longer limited to drip irrigation, furrow irrigation, flood irrigation, spraying, and other conventional methods in order to maximize the benefits of BS resources. The application of BS in combination with chemical fertilizer [79], solid organic fertilizer [186], biochar [187], duckweed [188], as well as pesticide [189,190,191], has become desirable.

A study found that 150 mL/m^2^ BS plus 27 g/m^2^ urea combined application was beneficial to increase the yield of dandelion and increase the content of vitamin C, nitrate, and soluble protein [192]. In another related study, the combined application of pig manure BS and earthworm fertilizer significantly improved the yield and quality of flat peach fruit [90]. The combination of BS and biochar increased the mass fraction of soil water stable aggregates [193]. Moreover, the combined application of BS and biochar for 3 years effectively increased the mass fraction of soil water-stable aggregates with a particle size of >0.25 mm, which is 13.0–36.3% higher than that of the control [187]. When the amount of biochar is constant (12 t/hm^2^), with the change in BS ratio, soil water-stable aggregate organic carbon and soil organic matter showed an increasing trend with a gradual increase in BS concentration [194,195]. When the ratio of BS is constant, with the increase in biochar dosage, the soil quality gradually decreased [195]. The application of 6% biochar from biogas residues can significantly reduce the leaching amount of BS nitrogen in lime-soil. The leaching amounts of total nitrogen, ammonium nitrogen, nitrate nitrogen, and nitrite nitrogen were reduced by 12.06, 11.82, 1.14, and 0.103 kg/hm^2^, and the declines were 35.9%, 53.0%, 25.5%, and 23.3%, respectively [196].

Furthermore, the combined application of BS concentrate and chemical fertilizer increased the rapeseed yield by 9.7% [197]. Also, the combined application of BS concentrates with chemical fertilizer and chicken manure significantly improved the quality of tomato with the contents of vitamin C and sugar increased by 9.4% and 49.5%, respectively, while the nitrite content was decreased by 27.1% [198]. Similarly, the application of BS concentrated with amino acid formula fertilizer increased banana yield by 4.1%, banana fruit protein by 10.7%, and vitamin C by 3.3%. It also increased the pH value of acidic soil and soil organic matter content by 3.0% and 3.9%, respectively [67], while foliar spraying of amino acid BS increased the pulp hardness and soluble solid content of cantaloupe by 13.9% and 7.7%, respectively [95]. Similarly, the addition of nutrients to BS concentrate, combined with berberine showed a strong inhibitory effect on tomato botrytis cinerea [199]. A related study using chicken manure BS concentrate diluted 300–500 times and mixed with pyridaben reduced the amount of pesticide application by 10–20%, which not only achieves the purpose of pest control, but also delays the enhancement of pest resistance, as well as reduced the cost of pest control [200]. Also, concentrated chicken manure BS and flonicamid were used in a combination to control apple yellow aphid; when the pesticides usage were reduced by 10% to 20%, the control effect was better than or equal to the conventional dosage of flonicamid [201].

### 3.4. Challenges of Using Biogas Slurry in Farmland

#### 3.4.1. Water Environment

Surface runoff: The nitrogen and phosphorus nutrients in BS are mostly and readily available nutrients. When the nitrogen and phosphorus nutrients provided in the BS exceed the needs of crop growth, they will continue to accumulate in the soil, in the face of heavy rainfall, improper irrigation, and poor drainage system. This can easily cause nutrient loss, leading to the eutrophication of rivers and lakes [188]. A case study is the paddy field engineering approach to BS valorization; the use of paddy field engineering for digesting BS is different from the utilization of paddy field fertilizer [40]. The first 3 days after irrigation is a critical period for the digestion of BS in paddy fields [202,203,204], and it is also a critical period for controlling nitrogen loss in paddy field runoff [40,205,206]. The risk of nutrient loss faced by BS application in a paddy field increased with the increase in application years. This will not only be reflected in the soil fertility index and nutrient accumulation rate, but also reflected in a lower soil ratio of C:P and N:P [71]. There might be a risk of nutrient loss in paddy soil with continuous application of BS for 4 years. To reduce the risk, the construction of farmland infrastructure such as a farmland ecological interception ditch system should be strengthened, and a series of agronomic, biological, and other supporting measures such as fertilizer and water management should be taken.

Downward leaching: The application of BS increases the nutrients such as nitrogen and phosphorus in the soil [103], these nutrients might also be leached downward with the BS and rainwater, posing a potential threat to farmland health, and even causing secondary environmental pollution [171]. For example, when BS was applied in vegetable fields, phosphorus accumulated, while nitrogen leaching loss occurred in surface soil [207]. A 3-year field trial of the mixed application of BS and irrigation water during the wheat–maize rotation in the North China Plain found using mild concentrated BS instead of chemical fertilizers was a reasonable method to ensure high crop yield, high nitrogen usage efficiency and reduction in nitrate leaching losses [208]. The leaching amount of ammonium nitrogen produced by the application of BS in the autumn fallow period was related to the growing season of crops, the amount of BS, and the BS application method. With the increase in the BS application rate, the risk of ammonium nitrogen leaching increases. Meanwhile, BS injection treatment increases the leaching potential of ammonium nitrogen compared to the spray treatment. Field experiments for three consecutive years showed that the content of ammonium nitrogen in soils of each BS nitrogen-treated soil was lower than that of no nitrogen application, indicating that no leaching of ammonium nitrogen occurred.

#### 3.4.2. Soil Environment

Heavy metals and antibiotic residues: Due to the use of different chemical compounds in animal feeds with various chemical additives and antibiotics being abused, the presence of heavy metals and antibiotics in livestock and poultry is low. Notwithstanding this, a considerable part of heavy metal and antibiotic pollutants are left in the feces [209,210]. During the anaerobic fermentation process, the heavy metals and antibiotic pollutants enriched in the manure will also remain in the BS [15,211,212]. Although the content is very low, if BS is applied blindly for a long time, it will introduce the risk of excessive heavy metals and residual antibiotic pollutants in farmland, which will destroy the farmland ecosystem causing food security problems [69,213]. Detection and analysis of soil and crops after application of BS showed that Cd, As, Pb, Ni, Cr, Cu, and Zn accumulated in different degrees in soil, and Cd, As, Pb, Ni, Cr, and Zn were enriched in different degrees in crops [214,215,216]. Long-term or high-concentration application of BS fermented with pig manure, chicken manure, cow manure, and other raw materials will lead to the accumulation of heavy metals in the soil. However, due to the different types of livestock, types of feeds, and amounts of feed, the content of heavy metals in the BS will be different. Therefore, reasonable control of the BS amount will reduce the pollution of heavy metals in the soil [217]. A standard control of feed additives to prevent the input of heavy metals and antibiotics for safe utilization of livestock and poultry manure BS. In this regard, the Chinese government has formulated guidelines on the safe use of additives in feed and also issued a policy on the complete prohibition of antibiotics in Chinese feed from 2020.

Secondary salinization: As a renewable water resource, BS could provide a large amount of nitrogen and phosphorus nutrients while solving the water shortage in arid and semi-arid areas. Therefore, BS irrigation is one of the important ways of recycling and using wastes at present [218,219]. However, BS also contains excess sodium ions, potassium ions and bicarbonate ions. Improper irrigation may cause excessive accumulation of soil salt, leading to soil salinization and potential pollution risks to farmland soil. For instance, in the vegetable planting base of Yining, Xinjiang, China, five consecutive years of BS irrigation show that with the increase in years of BS irrigation, salt accumulated in the farmland soil, resulting in secondary salinization of the soil [57]. Also, the nutrient and salinity accumulation in the soil of a protected vegetable field treated with pig manure BS for 0, 1, 3, 5, and 7 years was investigated. The results showed that the available nitrogen, organic matter, total copper, total zinc, and electrical conductivity in the soil showed an increasing trend year after year. After 7 years, each index was 3.4, 1.5, 3.3, 1.3, 3.9, 1.88, and 4.74 times that in the soil without the application of pig manure BS, respectively. This led to the rapid accumulation of salt while simultaneously increasing soil nutrients that put the soil at a risk of soil pollution [220]. In another study, the BS microbial fertilizer obtained from anaerobic digestion of kitchen waste was applied to single-season rice and winter wheat, it was observed that the water-soluble total salt and chloride ion in winter wheat and rice soil showed weak accumulation [221]. The field experiment used a Na^+^ concentration of about 35 mmol/L BS to irrigate oil sunflower for a long time. Under the low irrigation amount (150 m^3^/hm^2^), the agronomic profile of oil sunflower did not change much, and the difference ratio of K^+^/Na^+^ was not significant. But under high irrigation (600 m^3^/hm^2^), various agronomic indicators of oil sunflower growth were inhibited, and the K^+^/Na^+^ ratio of each tissue decreased by 57–88%. Alkaline salt could damage the ion homeostasis of oil sunflower to a greater extent, affecting the germination and growth of oil sunflower [222]. BS is used on farmland as regenerated water resources for the irrigation of farmland. Therefore, when applying BS in agriculture, consideration should be given to controlling the amount of salt accumulation in the soil from the source, thus reducing the risk of soil salinization.

#### 3.4.3. Nitrogen in BS and the Atmospheric Environment

More than 70% of the nitrogen in the BS in the form of NH_4_^+^-N [223,224] can be directly decomposed into gaseous ammonia and volatilized after being applied to the soil [19]. Therefore, the application of BS will increase the amount of soil ammonia volatilization [225,226,227] and become the most important contributor to the loss of NH_3_-N after returning to the field [228,229]. The amount of BS, time of application, temperature, and application method will all affect the amount of ammonia volatilization. The larger the amount of BS applied in the field, the larger the amount of ammonia volatilization [110]. For instance, in a pot experiment, it was found that the total amount of ammonia volatilization using conventional chemical fertilizer treatment was 77.0 kg/hm^2^, while the amount of ammonia volatilization from 100% BS treatment and 75% BS plus 25% pig manure organic fertilizer treatment was higher, at 120.7 kg/hm^2^ and 88.0 kg/hm^2^, respectively [186]. Moreover, when BS was applied at low temperature in the autumn fallow period, the peak value of ammonia volatilization in spraying and injection treatment was 0.22 kg/(hm^2^·d) and 0.65 kg/(hm^2^·d), respectively [41,110,182]. Furthermore, a study by Jin H et al. [225] pointed out that more than 58% of ammonia volatilization loss is related to environmental temperature after BS has been applied to soil. When the temperature during BS application is higher, the amount of ammonia volatilization increases significantly. Studies also show that after the application of BS, the volatilization of ammonia was higher the in the previous week, after which the volatilization of ammonia gradually decreased and thereafter became stable [204,230]. In addition to the influence of environmental temperature on ammonia volatilization, there is a positive correlation between the ammonia volatilization flux and the NH_4_^+^-N concentration of BS in the field surface water [231,232,233].

The reason for the increase in ammonia volatilization rate is not only due to the high NH_4_^+^-N content, environmental temperature, and presence of surface water in the soil [225], but also because the BS contains a large amount of soluble organic carbon, which can stimulate the mineralization of soil organic nitrogen [228,234]. Usually the amount of ammonia volatilization loss correlates with the proportion of BS application, and the proportion of nitrogen loss caused by ammonia volatilization [230]. For instance, ammonia volatilization in paddy field treated with BS was 42.2–72.0% of total nitrogen loss [227]. However, in the conventional storage of BS, 25–35% of N was lost in the form of NH_3_-N [235,236]

#### 3.4.4. BS Application and Crop Safety

Appropriate BS concentration and dosage can promote plant growth and development, as well as improve yield and quality. However, when the concentration and dosage are improperly applied, the growth and development of crops will be affected. Ammonia, phenols, hydrogen sulfide, and high chemical oxygen demand (COD) in BS may cause the anoxic death of plant roots as well as the slow growth development of plants [237]. The COD content is the key limiting factor affecting the use of BS in farmland. A low amount of COD (1566 kg/hm^2^) in BS promotes seedling growth, accelerates the peak supply of soil available phosphorus, while a high amount of COD (3132 kg/hm^2^) inhibits seedling growth and delays the peak supply of soil available phosphorus. The optimal application safety threshold of COD is 1102–1442 kg/hm^2^ and the maximum application safety threshold is 2208–2884 kg/hm^2^. This factor needs to be taken into consideration for farmland BS safe usage and efficiency [238]. Moreover, when the concentration of ammonium nitrogen and lactic acid in BS is higher than 336 and 61 mg/L, respectively, it could produce phytotoxicity to seed germination [239]. Excessive application of BS increases NH_4_^+^-N concentration and electric conductivity (EC) value in soil solution, resulting in inhibited seedling growth, decreased plant height, and increased root yellowing rate. The maximum safe absorption threshold of NH_4_^+^-N in BS water mixture by seedlings was 314.0 mg/L. It is also found that the EC value of BS increases with the increase in BS concentration, and a possible synergistic effect between EC and NH_4_^+^-N concentration still needs to be further studied [240].

In addition, the application of BS can increase the content of heavy metals in plants. For instance, Shao W et al. [75] observed a varying concentration of Hg, As, Cr, and Pb in rice straw. Also, the application of high-concentration pig farm BS (l.8 × 10^5^ kg/hm^2^) significantly increased the copper content in lettuce and in Chinese cabbage, but it was lower than the limit range stipulated by the national food hygiene standards [241,242]. When the BS contains four times the nitrogen equivalent, it will increase the excessive enrichment of Cu and Zn elements, which will have a negative effect on crop growth, reducing the yield and quality of plants such as corn [64]. A 5-year irrigation experiment using pig manure BS in rice–wheat rotation field in Dongtai, Jiangsu, China, showed that Zn in the grains of wheat and rice increased by 24% and 16%, respectively, compared with the control [135].

With the continuous deepening of scientific research, some scholars have paid attention to the presence of contaminants of emerging concern (CEC) in BS (e.g., hormones, antibiotics, etc.) [243,244]. Hormones and antibiotics [245], as well as heavy metals, are used in excess during livestock and poultry breeding, resulting in retention in excreta. When excreta are used for biogas plants, the heavy metals, antibiotics, and hormones may remain in the BS. Although the residual concentration is very tiny, long-term application in farmland may still pose a cumulative risk. In fact, what needs to be paid attention to is whether the government is in position to supervise the implementation of limit standards related to CEC, strictly following the limit standards to breeding livestock and poultry, which would address the anxiety-based attention at its source. On the other hand, it is worth studying whether the residual CEC substances in BS have the ability to accumulate for a long time in farmland soil, and whether they can pose a threat to human health through plant enrichment. However, it is not an excuse to hinder the utilization of biogas slurry in farmland for currently, applying BS to farmland is the most economical and practical effective method to dispose of BS.

## 4. Conclusions and Prospects

The advantages of utilizing biogas slurry on farmlands has been elucidated. Current research mostly focuses on the advantages of biogas slurry utilization but there is a lack of in-depth research on the underlying mechanisms. There is a lack of research on the application methods, equipment, relevant engineering measures, parameters, and environmental benefits assessment of base fertilizer and topdressing.

There is also a lack of systematic understanding of the concentration, time, temperature, and operational precautions during BS application for different plants and farmlands.

In addition, due to the lack of research on the underlying mechanisms in BS utilization, the application effects in different regions is unsatisfactory, which could limit biogas slurry utilization in farmland.

There are numerous advantages of BS application as listed in the current review. However, farmland application of BS needs to be further studied to: (1) establish fitting models for different components of BS adsorbed by soil, maximum adsorption capacity of different types of soil, and environmental factors affecting soil adsorption; (2) explore the transformation and characteristics of BS components in its farmland application, the time cycle of safe application, as well as the assessment of associated potential risks; (3) develop combined technologies, equipment, and engineering measures to increase BS high value utilization; and (4) analyze the mechanism as well as microecological mechanism of BS digested in farmland to improve soil fertility and productivity.

## Figures and Tables

**Table 1 microorganisms-11-02675-t001:** The nutrient element, heavy metals, antibiotics, and other residual substances content of different types of biogas slurry [9,10,15,43,44].

Element in Biogas Slurry	PM		DM		CM		MM	
	Range	Average	Range	Average	Range	Average	Range	Average
pH	4.23–9.20	7.52	6.10–9.20	7.75	6.77–8.50	7.80	6.15–8.20	7.37
TN (mg/L)	0.80–7280.00	1166.71	32.00–6500.00	1488.59	400.00–5700.00	3226.13	0.04–5900.00	1369.31
TP (mg/L)	0.54–2220.50	291.60	10.00–3700.00	561.67	49.00–4650.00	959.71	0.03–3900.00	665.90
TK (mg/L)	0.33–8880.00	1144.26	11.00–9650.00	1679.10	390.00–4400.00	2858.31	0.12–3200.00	1240.21
NH_4_^+^-N (mg/L)	66.53–1800.00	597.53	80.35–1098.00	493.47	ND	ND	250.50–787.80	519.15
NO_3_^−^-N (mg/L)	0.19–472.16	67.84	0.70–223.70	71.53	ND	ND	ND	ND
DP (mg/L)	0.16–1730.00	261.40	80.00–1860.00	416.88	ND	ND	0.16–201.10	76.68
DK (mg/L)	0.86–5010.00	986.47	263.20–2500.00	1418.33	ND	ND	0.84–2316.70	764.73
Hg (mg/L)	0–0.167	0.028	0–0.119	0.024	0–0.054	0.014		
Cd (mg/L)	0–7.51	0.126	0–0.190	0.039	0–4.3	0.367		
As (mg/L)	0–13	0.868	0.001–4.576	0.235	0.01–5.21	0.548		
Pb (mg/L)	0–36.07	0.710	0.008–1.056	0.199	0–2.430	0.345		
Cr (mg/L)	0–24.18	0.657	0–3.146	0.301	0.001–10.18	1.085		
Ni (mg/L)	0–5.85	0.317	0.027–0.063	0.045	0.088–0.55	0.281		
Cu (mg/L)	0–99	4.50	0.02–30.03	2.63	0–2.12	0.78		
Zn (mg/L)	0–205.43	9.11	0.1–68.15	8.31	0–13.94	4.06		
Cl (mg/L)	150–3647.5	917.1	850.5–963	906.8	540–1087	813.5		
Na (mg/L)	88.5–559	287.1	994.45	994.45	172.29	172.29		
Se (mg/L)	0–0.232	0.049	0.002–0.022	0.012	0.011	0.011		
Mn (mg/L)	0–50.8	6.815	0.231–124.6	61.092	0–50.8	7.534		
Fe (mg/L)	0.0014–6.05	2.505	0.0084–48.3	18.56	0.0054–13.3	4.962		
Mg (mg/L)	0.0057–253.34	109.6	0.352–553	225.06	0.0109–89.46	32.82		
Ca (mg/L)	0.0042–264	81.65	0.785–769	280.8	0.0426–96.1	61.14		
Oxytetracycline (mg/L)	0.0001–0.994	0.1456	0.5748	0.5748	0.0759–0.4007	0.2383		
Tetracycline (mg/L)	0–0.9821	0.0296	0.0208–0.5608	0.2908	0.0289–12.862	4.3106		
Chloromycin (mg/L)	0.0002–0.642	0.0415	ND	ND	ND	ND		
Norfloxacin (mg/L)	0–0.204	0.0191	0.0054–0.1189	0.0641	0.056–0.2048	0.1065		
Ciprofloxacin (mg/L)	0.0002–0.0513	0.0052	0.016–0.0227	0.0183	0.005–0.0071	0.0058		
Enrofloxacin (mg/L)	0–0.1513	0.0108	0.0058–0.089	0.0520	0.0073–0.0676	0.0519		

ND: no data; PM: biogas slurry using pig manure as fermentation raw material; DM: biogas slurry using cow dung as fermentation raw material; CM: biogas slurry using chicken manure as fermentation raw material; MM: biogas slurry using two or more of straw, human excrement, pig manure, cow dung, chicken manure, and other household waste mixtures as fermentation raw material; TN: total nitrogen; TP: total phosphorus; TK: total potassium. NH_4_^+^-N: ammonium nitrogen; NO_3_^−^-N: nitrate nitrogen; DP: available phosphorus; DK: available potassium.

**Table 2 microorganisms-11-02675-t002:** Contents of amino acids, plant hormones, and B vitamins in biogas slurry [9,12].

Amino Acids	Contents (mg/L)	Amino Acids	Contents (mg/L)	Plant Hormones	Contents (μg/L)	B Vitamins	Contents (mg/L)
Cysteine	2.92	Arginine	0.63	Indole acetic acid (IAA)	332	B1	0.089
Serine	2.07	Proline	0.58	Gibberellin (GA4)	0.857	B2	0.022
Threonine	1.41	Valine	0.56	Gibberellin (GA19)	1.47	B6	0.530
Lysine	1.05	Leucine	0.45	Gibberellin (GA53)	0.271	B11	0.078
Glycine	1.01	Methionine	0.36	Cytokinin (iPR)	0.00194	B12	0.009
Tyrosine	0.88	Alanine	0.36	8-hydroxy-3,4-dihydroquinoline-2-ketone	737.5		
Aspartic acid	0.76	Phenylalanine	0.33	3,4-dihydroquinoline-2-ketone	177.5		
Isoleucine	0.75	Glutamate	0.31				
Histidine	0.63						

**Table 3 microorganisms-11-02675-t003:** Yield-increasing effect of applying biogas slurry to farmland.

Slurry Type	Crops	Production Increase Range		References
		Comparison with Conventional Chemical Fertilizer	Comparison with No Fertilization	
PM	Rice	0.2–20.4%	1.0–102.5%	[61,74,75,76,77,78,79,80]
	Wheat	2.9–22.4%	97.1–217.5%	[74,76]
	Corn	0.6%	5.6–13.2%	[81,82]
	Barley	1.1–2.0%	31.9–111.9%	[83]
	Watermelon	0.2–24.9%	ND	[84,85,86]
	Pear	12.0%	3.1–7.4%	[87,88]
	Grape	ND	10.7–18.3%	[89]
	Peach	ND	9.7–43.7%	[90]
	Cabbage	ND	75.4–133.9%	[91]
	Tea	ND	9.3–93.4%	[92]
DM	Corn	ND	59.2–81.7%	[70]
	Melon	8.8–32.2%	8.6–33.0%	[93,94,95]
	Grape	30.0–170.0%	ND	[96]
CM	Corn	9.0–26.2%	12.9–107.7%	[63,64]
	Apple	2.0–3.5%	42.8–67.0%	[97]
	Leafy vegetables	9.1–45.1%	45.1%	[98,99,100]
MM	Rice	4.6–7.7%	4.1%	[101,102]
	Wheat	ND	28.2–71.1%	[103]
	Tomato	0.49–21.59%	15.6–39.8%	[104,105]
	Cabbage	8.5–41.2%	ND	[106]
	Tangerine	11.8–24.8%	32.4–56.4%	[107,108]

ND: no data; PM: biogas slurry using pig manure as fermentation raw material; DM: biogas slurry using cow dung as fermentation raw material; CM: biogas slurry using chicken manure as fermentation raw material; MM: biogas slurry using two or more of straw, human excrement, pig manure, cow dung, chicken manure, and other household waste mixtures as fermentation raw material; TN: total nitrogen; TP: total phosphorus; TK: total potassium. NH_4_^+^-N: ammonium nitrogen; NO_3_^−^-N: nitrate nitrogen; DP: available phosphorus; DK: available potassium.

**Table 4 microorganisms-11-02675-t004:** Improvement in soil enzyme activity via applying biogas slurry in farmland.

Soil Type	Crops	Slurry Type	Total Nitrogen Consumption of Slurry (kg/hm^2^)	Years	Improvement Range of Soil Enzyme Activity		References
					Comparison with No Fertilization	Comparison with Conventional Chemical Fertilizer	
Yellow loamy paddy soil	Rice–rape	PM	157.5–694.9	3	Urease 31.8–74.6%, catalase 4.4–85.1%, sucrase 30.4–228.6%	Urease 21.7–61.1%, catalase 19–111%, sucrase 45.4–266.4%	[140]
Retention paddy soil	Rice–wheat	PM	210.3–540.9	3	Urease 30.5–79.5%, protease 19.1–41.4%, phosphatase 11.3–29.7%, catalase 6.4–40.1%, sucrase 0.2–39.3%, amylase 53.1–161.4%, cellulase 15.8–104.8%, lactase 30.1–65.2%	Urease 26.7–74.8%, protease 9.1–43.3%, phosphatase 7.3–8.8%, catalase 14.9–30.2%, sucrase 3–49.4%, amylase 28–145.7%, cellulase 38.5–48.3%, lactase 13.3–37.1%	[76]
Medium loam	Wheat	PM	60–180	1	Urease 57.5–72.5%, protease 31.7–62.6%	Urease 10.5–21.1%, protease 6–30.1%	[149]
Red soil	Peanut	PM	36–240	2	Urease 16.2–62.3%, dehydrogenase 48.6–133.1%	Urease 11.3–24.1%, dehydrogenase 33.8–120.5%	[144]
Aeolian sandy soil	Grape	DM	190–1160	2	ND	Urease 41–113.8%, phosphatase 32.4–106.4%, sucrase 62.7–98%	[96]
ND	Cucumber	(P+C)M	37.5–150	1	Polyphenol oxidase 13.49–14.75%, cellulase 68.7–71.9%, chitinase 41.0–57.5%,	ND	[150]
ND	Cabbage	(P+S)M	504–675	3	ND	Urease 2.4%, protease 95.4–139.7%, phosphatase 50.5–137.6%, invertase 55.7–64.0%	[151]
Paddy soil	Citrus	(P+D)M	450	4	ND	Urease 53.8–100.0%, protease 23.1–100.0%, phosphatase 20.2–42.3%, catalase 107.0–127.5%, dehydrogenase 36.6–96.0%, sucrase 47.4–111.8%	[108]

ND: no data; PM: biogas slurry using pig manure as fermentation raw material; DM: biogas slurry using cow dung as fermentation raw material; (P+C)M: biogas slurry using pig manure and chicken manure as fermentation raw material; (P+S)M: biogas slurry using pig manure and straw as fermentation raw material; (P+D)M: biogas slurry using pig manure and cow dung as fermentation raw material.

## Data Availability

Data will be made available on request.

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
