# Peer review of "Developments and Prospects of Farmland Application of Biogas Slurry in China—A Review"

_microorganisms, 2023, doi:10.3390/microorganisms11112675_

Round 1

Reviewer 1 Report

Comments and Suggestions for Authors

Major comments

The work’s main findings, implications, and research directions are missing in the abstract.

The quality of the graphical abstract should be improved.

What about the current scenario of biogas slurry destinations around the globe? Numerical data on this aspect need to be included and compared.

The study’s novelty is unclear and should be elaborated on based on what was already published.

Tables should be self-explanatory. Please add the references in the tables.

Conclusions and prospects should be mixed as a single section. Also, conclusions should be shorter. Please summarize the main findings, their implications, and research directions.

It needs to be clarified how BS can be valorized. Elaborate more on this aspect.

The topic is exciting, and the paper could foster future studies. Authors are advised to copyedit the work for readability improvements.

Comments on the Quality of English Language

The manuscript's readability should be improved.

Author Response

Response to Reviewer 1 Major Comments

Point 1: The work’s main findings, implications, and research directions are missing in the abstract.

Response 1: We greatly appreciate your positive comments. We have revised the abstract, as follows:

Biogas slurry (BS) is an attractive agricultural waste resource which can be used to regulate soil microbial communities, enhance nutrient absorption capacity of crops, promote plant-soil interactions, and consequently, increase crop productivity. Presently, BS discharge is not environmentally friendly. It is therefore necessary to explore alternative efficient utilization of BS. The use of BS as fertilizer meets the requirements for sustainable and eco-friendly development in agriculture but this has not been fully actualized. Hence, this paper reviewed the advantages of using BS in farmland as soil fertilization for the improvement of crop production and quality. This review also highlighted the potential of BS for the prevention and control of soil acidifica-tion, salinization as well as improve microbial structure and soil enzyme activity. Moreover, this review reports on the current techniques, application methods, relevant engineering measures, environmental benefits, challenges and prospects associated with BS utilization. Lastly, additional research efforts require for optimal utilization of BS in farmlands were elucidated.

Point 2: The quality of the graphical abstract should be improved.

Response 2: Thank you for your suggestion. We have improved the graphical abstract, as follows:

Point 3: What about the current scenario of biogas slurry destinations around the globe? Numerical data on this aspect need to be included and compared.

Response 3: We have added the words “in China” in the Title for the manuscript is focused on cases from China.

Point 4: The study’s novelty is unclear and should be elaborated on based on what was already published.

Response 4: This paper reviewed the advantages, approaches, application methods, challenges and prospects associated with BS utilization, using systematic approach overview of online sources, peer-reviewed articles, and published books. Aimed to show directions of farmland application of BS needs to be further studied, and to provide additional reference for new BS utilization development towards eco-friendly and circular agricultural economy.

Point 5: Tables should be self-explanatory. Please add the references in the tables.

Response 5: Revised as suggested.

Point 6: Conclusions and prospects should be mixed as a single section. Also, conclusions should be shorter. Please summarize the main findings, their implications, and research directions.

Response 6: Revised as suggested.

Point 7: It needs to be clarified how BS can be valorized. Elaborate more on this aspect.

Response 7: We greatly appreciate your insightful suggestions. The agricultural application of biogas slurry has strong public welfare and requires government financial support. Developing low-cost and efficient digestion technologies for biogas slurry, while also achieving additional high economic benefits, is one of the main directions of current research. We also reviewed new research directions for improving the byproduct value of biogas slurry in the article, such as biogas slurry concentration technology and product application, and biogas slurry aquaculture functional microalgae, but none of them have been widely promoted and applied in actual production practice.

Point 8: The topic is exciting, and the paper could foster future studies. Authors are advised to copyedit the work for readability improvements.

Response 8: Revised as suggested.

Reviewer 2 Report

Comments and Suggestions for Authors

The manuscript presents an interesting issue concerning farmland application of biogas slurry but needs substantial improvement.

For example, the structure of the manuscript must be changed because it has too many subchapters, some of which have few references. For example, in chapter “3.4 Challenges of using biogas slurry in farmland” consider dividing it into just 3 subchapters (water, soil and air).

The entire manuscript is focused on cases from China, so it should be changed, with the introduction of references from other countries or changing the title to "Developments and prospects of farmland application of biogas slurry in China - A review”.

Page 2. Line 79. “Through the search and analysis of over 750 documents on biogas slurry release to the field in China in the past 20 years from January 2000 to December 2019, the nutrient composition of BS are shown in Table 1 [10,43].” There are several papers concerning the composition of BS, just 2 references are not enough. Please consider presenting the parameters in Tables 1 to 3 in a single table.

Comments on the Quality of English Language

Please revise the English. Example Page 2, Line 81, “composition of BS are shown” change to “composition of BS is shown”

Author Response

Response to Reviewer 2 Major Comments

Point 1: The manuscript presents an interesting issue concerning farmland application of biogas slurry but needs substantial improvement.

For example, the structure of the manuscript must be changed because it has too many subchapters, some of which have few references. For example, in chapter “3.4 Challenges of using biogas slurry in farmland” consider dividing it into just 3 subchapters (water, soil and air).

Response 1: We greatly appreciate your positive comments. We have changed as suggested.

Point 2: The entire manuscript is focused on cases from China, so it should be changed, with the introduction of references from other countries or changing the title to "Developments and prospects of farmland application of biogas slurry in China - A review”.

Response 2: Thank you for your suggestion. We have changed the title to "Developments and prospects of farmland application of biogas slurry in China - A review”.

Point 3: Page 2. Line 79. “Through the search and analysis of over 750 documents on biogas slurry release to the field in China in the past 20 years from January 2000 to December 2019, the nutrient composition of BS are shown in Table 1 [10,43].” There are several papers concerning the composition of BS, just 2 references are not enough. Please consider presenting the parameters in Tables 1 to 3 in a single table.

Response 3: Revised as suggested.

Point 4: Please revise the English. Example Page 2, Line 81, “composition of BS are shown” change to “composition of BS is shown”.

Response 4: Revised as suggested.

Round 2

Reviewer 1 Report

Comments and Suggestions for Authors

Dear authors,

Responses to points 3 and 4 are not satisfactory. This review expected a complement to the raised issues, which are critical considering the scope of this paper. These parst of the article must be extended and discussed further.

The status of digestate applications in other countries needs to be discussed, even though the paper focuses on China.

Section 2: The authors brought to attention the presence of contaminants of emerging concern (CEC) in BS (e.g., hormones, antibiotics, etc). How could this be a barrier to BS usage in crops, and how such an issue could be solved? This must be discussed. From my knowledge, CECs in digestate tend to limit their application in soil, as mentioned in https://doi.org/10.1016/j.jclepro.2021.129056

“digestate application to the agricultural fields will be more and more discouraged due to the increasing presence in digested sludge of heavy metals (Khakbaz et al., 2020) and emerging pollutants, such as microplastics (van den Berg et al., 2020) and antibiotic resistance genes (Urra et al., 2019).”

Once again, the topic is exciting, and the paper could foster future studies. However, authors should be willing to improve the article content as suggested.

Comments on the Quality of English Language

Comments were addressed to the authors. Paper must be copyedited before publication.

Author Response

Point 1: Responses to points 3 and 4 are not satisfactory. This review expected a complement to the raised issues, which are critical considering the scope of this paper. These parst of the article must be extended and discussed further.

Response 1: We greatly appreciate your insightful comments.

Undoubtedly, following your suggestions to make revisions will greatly improve the quality of this article, because the data on the global annual production of biogas slurry has not been publicly reported. The production of biogas slurry cannot be separated from biogas projects. European countries, such as Germany, have developed biogas plants earlier and quickly reached its peak in 2008, but were forced to stop operation soon afterwards. Asian countries, such as India, is second only to China, with more biogas projects development. Countries in the Americas and Africa also have biogas projects. However the degree of data disclosure of the biogas projects and biogas slurry is limited, making it difficult to collect relevant real data.

Point 2: The status of digestate applications in other countries needs to be discussed, even though the paper focuses on China.

Response 2: The proposal to increase the discussion on the utilization of biogas slurry in other countries is very good. We also believe that it is necessary to write an review article about the utilization of biogas slurry from a global perspective. It could serve as a strong support to the work if this manuscript can be published which based on the situation of biogas slurry in China.

Point 3: Section 2: The authors brought to attention the presence of contaminants of emerging concern (CEC) in BS (e.g., hormones, antibiotics, etc). How could this be a barrier to BS usage in crops, and how such an issue could be solved? This must be discussed. From my knowledge, CECs in digestate tend to limit their application in soil, as mentioned in https://doi.org/10.1016/j.jclepro.2021.129056.

“digestate application to the agricultural fields will be more and more discouraged due to the increasing presence in digested sludge of heavy metals (Khakbaz et al., 2020) and emerging pollutants, such as microplastics (van den Berg et al., 2020) and antibiotic resistance genes (Urra et al., 2019).”

Response 3: Thank you for your suggestion. Revised as suggested in red.

With the continuous deepening of scientific research, some scholars have raised these issues and conducted relevant experimental analysis and verification, which is worthy of attention. Heavy metals, antibiotics, hormone are used in excess during livestock and poultry breeding, resulting in retention in excreta. When excreta are used for biogas plants, the heavy metals, antibiotics, hormone may remain in the biogas slurry. Although the residual amount is very small, long-term application in farmland still poses a cumulative risk. What needs to be paid attention to is whether the government is in place in supervising the implementation of limit standards related to aquaculture, strictly following the limit standards to breeding livestock and poultry, which solves this anxiety problem from the source. On the other hand, it is worth studying whether the residual contaminants of emerging concern substances in biogas slurry have the ability to accumulate for a long time in farmland soil, and whether they can pose a threat to human health through plant enrichment. However, it is not an excuse to hinder the utilization of biogas slurry in farmland for currently, applying biogas slurry to farmland is the most economical and practical effective method to disposal biogas slurry.